# Learning Safety Constraints From Demonstration Using One-Class Decision Trees

**Mattijs Baert, Sam Leroux, Pieter Simoens**

IDLab, Department of Information Technology at Ghent University - imec
Technologiepark 126
B-9052 Ghent, Belgium
{mattijs.baert, sam.leroux, pieter.simoens}@ugent.be

## Abstract

The alignment of autonomous agents with human values is a pivotal challenge when deploying these agents within physical environments, where safety is an important concern. However, defining the agent's objective as a reward and/or cost function is inherently complex and prone to human errors. In response to this challenge, we present a novel approach that leverages one-class decision trees to facilitate learning from expert demonstrations. These decision trees provide a foundation for representing a set of constraints pertinent to the given environment as a logical formula in disjunctive normal form. The learned constraints are subsequently employed within an oracle constrained reinforcement learning framework, enabling the acquisition of a safe policy. In contrast to other methods, our approach offers an interpretable representation of the constraints, a vital feature in safety-critical environments. To validate the effectiveness of our proposed method, we conduct experiments in synthetic benchmark domains and a realistic driving environment.

## Introduction

Reinforcement Learning (RL) has made significant strides in training autonomous agents, but as these systems become more advanced, ensuring their safety and alignment with human intentions, often referred to as the alignment problem (Russell 2019), is becoming a critical concern. As a result, the objectives of autonomous agents extend beyond their primary task goals, also encompassing safety constraints, human values, legal regulations, and various other factors. For instance, an autonomous vehicle's objective is not only to get passengers quickly to their destination but also to abide by traffic laws and ensure passenger comfort. To address these challenges and enhance the safety of RL agents, Constrained Reinforcement Learning (CRL) methods have emerged. CRL is designed to obtain safe RL agents by incorporating constraints into the learning process. CRL methods help ensure that the RL agent operates within predefined boundaries, reducing the risk of harmful or unintended actions. However, one key limitation of CRL methods is that they assume the availability of a well-defined set of constraints. Specifying these constraints correctly and completely is a complex task on its own (Krakovna et al.

2020). It often requires a deep understanding of the environment, potential risks, and ethical considerations. Therefore, addressing the alignment problem in RL goes beyond the development of CRL methods; it also involves addressing the challenge of correctly specifying constraints as necessary. This research paper introduces a novel approach to reduce the need for manual specification of constraints by learning from expert demonstrations. First, a model of the expert behavior is learned as a one-class decision tree (Itani, Lecron, and Fortemps 2020). From this tree a logical formula in disjunctive normal form is extracted defining the constraints. Next, we demonstrate that the learned constraints can be used by a CRL method to learn a constraint abiding (i.e. safe) policy in synthetic and realistic environments. A notable advantage of this approach is the interpretability of both the learned expert model and the constraints themselves. By monitoring the evaluation-violation ratio of the constraints during training, we can refine the set of constraints after training, further enhancing interpretability. Additionally, as constraints are frequently shared across multiple agents and tasks, once they have been acquired, they can be seamlessly applied to other agents undertaking diverse tasks. This eliminates the need to separately learn constraints for each individual agent and task. We assess the effectiveness of the proposed approach through an evaluation on a series of synthetic benchmarks for safe RL (Ray, Achiam, and Amodei 2019) and in a real-world driving scenario (Krajewski et al. 2018).

## Background

### Markov Decision Process

A Markov decision process (MDP) is characterized by a state space $\mathcal{S}$, an action space $\mathcal{A}$, a discount factor $\gamma$ within the range [0,1], a transition distribution denoted as $p(s' \mid s, a)$, an initial state distribution represented by $\mathcal{I}(s)$, and a reward function $R : \mathcal{S} \times \mathcal{A} \mapsto [r_{\min}, r_{\max}]$. Within this framework, an agent interacts with the environment at discrete time steps, generating a sequence of state-action pairs known as a trajectory $\tau = ((s_0, a_0), ..., (s_{T-1}, a_{T-1}))$, where $T$ is the length of the trajectory. The cumulative reward of a trajectory is calculated as the sum of rewards, each discounted by a factor of $\gamma^t$, where t represents the time step: $R(\tau) = \sum_{t=0}^{T-1} \gamma^t R(s_t, a_t)$. At each discrete time step, the

agent's choice of action is governed by a policy $\pi$, which is a function that maps states from the state space $\mathcal{S}$ to a probability distribution over actions from the action space $\mathcal{A}$. The primary objective of forward reinforcement learning is to find a policy $\pi$ that maximizes the expected sum of discounted rewards, expressed as $J_r(\pi) = \mathbb{E}_{\tau \sim \pi} R(\tau)$.

## Constrained Reinforcement Learning

Constraints provide a natural and widely applicable means of specifying safety requirements in various contexts (Ray, Achiam, and Amodei 2019). Within the domain of Constrained Reinforcement Learning (CRL), the prevailing framework for problem modeling is the Constrained Markov Decision Process (CMDP) (Altman 1999). CMDPs extend the traditional Markov Decision Processes (MDPs) by introducing a non-negative bounded cost function, denoted as $C : \mathcal{S} \times \mathcal{A} \mapsto [c_{\min}, c_{\max}]$, and a budget parameter $\alpha \geq 0$. In this context, $C(s, a)$ quantifies the cost associated with taking action $a$ in state $s$, while the cumulative cost of a trajectory is defined as the summation of discounted costs: $C(\tau) = \sum_{t=0}^{T-1} \gamma^t C(s_t, a_t)$. The objective in CRL is to find an optimal policy that maximizes the expected sum of discounted rewards, as denoted by $J_r(\pi)$, while adhering to specific constraints defined by the cost function $C$. Formally, the optimal policy $\pi^*$ is obtained through the following optimization problem:

$$\pi^* = \arg\max_{\pi} J_r(\pi) \text{ s.t. } J_c(\pi) < \alpha. \tag{1}$$

Here, $J_c(\pi) = \mathbb{E}_{\tau \sim \pi} C(\tau)$ represents the expected sum of discounted costs, and $\alpha$ signifies a limit on the allowable cost.

## One-Class Decision Tree

One-class classification (OCC) is a machine learning paradigm that involves training a model to classify instances into a single well-defined class, treating all other data points as anomalies or outliers. One-class classification trees (OC-trees) (Itani, Lecron, and Fortemps 2020) provide an interpretable approach for addressing OCC problems. Building on kernel density estimation, OC-trees aim to represent target areas in the input space that describe the training data. We consider the training data as a set of $M$ instances $X = \{x_0, x_1, \ldots x_{M-1}\}$. Each instance is characterized by a k-dimensional feature vector, with $x_i^j$ representing the $j$-th feature of the $i$-th instance. We define $\chi \subset \mathbb{R}^k$ as a k-dimensional hyper-rectangle that encompasses all training instances. The primary objective is to partition the initial hyper-rectangle $\chi$ into distinct subspaces $\chi_t$, represented by tree nodes $t$, such that the learned subspaces encompass the training data. The sub-space $\chi_t$ associated with node $t$ is divided into one or more sub-spaces $\chi_{t_n} = \{x \in \chi_t : L_{t_n} \leq x^j \leq R_{t_n}\}$ with $L_{t_n}, R_{t_n} \in \mathbb{R}$, $n \in \{0, ..., N-1\}$ and $N$ the number of children of $t$. At a given node $t$, several steps are carried out for each dimension to determine the dimension $j$ which best cuts the data in multiple sub-spaces:

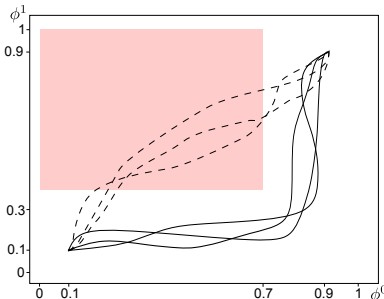

Figure 1: Simple navigation environment with a two dimensional feature space. The red region represents the ground truth constraints. The solid lines represent expert trajectories and the dashed line, trajectories a learning agent would take which is unaware of the constraints.

- Estimate the probability density function $\hat{f}_j(x)$ along dimension $j$ using kernel density estimation, based on the available training samples $x \in \chi_t$.
- Divide the subspace $\chi_t$ along dimension $j$ based on the modes of the estimated density function $\hat{f}_j(x)$ into $N$ intervals defined by their left and right bound $L_{t_n}$ and $R_{t_n}$. Note, that $N$ depends on the number of modes of $\hat{f}_j(x)$.
- Evaluate the quality of the division using a measure of *impurity*.

The dimension that yields the best purity score is selected to partition the subspace $\chi_t$.

## Method

Our approach builds upon the insight, as discussed by Lindner et al. (2023), that feature expectations of policies adhering to the genuine constraints form a convex set (i.e. safe set) within the feature vector space. If, for some policy, we can guarantee that the feature expectations are enclosed by the safe set then we can guarantee this policy is safe. Our method encompasses four key steps: firstly, a safe convex set is established through constructing an OC-tree; following this, constraints are extracted in the form of a logical formula; then, we employ a CRL method to train a policy that conforms to these constraints; and lastly, we propose an approach for pruning the learned formula after training.

### Learning a Safe Set

In this section, we discuss the process of acquiring a representation of safe behavior. We start with a collection of expert trajectories denoted as $\mathcal{T}$. We define a fixed mapping from state-action tuples to a bounded k-dimensional feature space as $\phi : \mathcal{S} \times \mathcal{A} \to \mathbb{R}^k$. Our goal is to learn a safe convex set in this feature space. To accomplish this, we construct a dataset $\mathcal{D}$ from the set of trajectories, defined as: $\mathcal{D} = \{\phi(s, a) : \forall (s, a) \in \tau; \forall \tau \in \mathcal{T}\}$. Training an OC-tree (Itani, Lecron, and Fortemps 2020) on the dataset $\mathcal{D}$ constructs a region in the feature space by taking the union of hyper-rectangles. Since hyper-rectangles themselves are

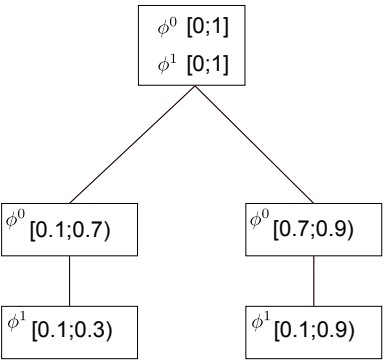

Figure 2: Expert behavior tree learned from trajectories presented in figure 1. The root of tree encompasses the complete feature space. Every other node defines an interval along one dimension such that the complete tree represents the safe set as multiple hyper-rectangles. This tree represents two rectangles in the two dimensional feature space.

convex, and the union operation is known to maintain convexity, it follows that the OC-tree effectively defines a convex set. To illustrate this concept, let's consider a simplified environment characterized by a two-dimensional feature space. In this scenario, the first dimension, denoted as $\phi^0$, corresponds to the agent's x-coordinate, while the second dimension, $\phi^1$, represents the agent's y-coordinate. Figure 1 visually represents this environment along with three expert trajectories depicted as solid lines. The agent is tasked with a straightforward navigation assignment, starting from an initial feature vector of $[0.1, 0.1]$ and aiming to reach a goal vector of $[0.9, 0.9]$. The red rectangle in Figure 1 represents a constrained area within the feature space. Figure 2 showcases the OC-tree that has been learned from the provided trajectories, which we refer to as the expert tree or expert model. In this specific instance, the expert behavior is enclosed by two rectangles within the feature space.

## Formula Extraction

Our objective is to acquire a logic formula $\varphi$ in Disjunctive Normal Form (DNF) that captures the constraints governing the expert's behavior. $\varphi$ should evaluate true for the portion of the feature space not covered by the safe set represented by the expert tree. For a tree defined by its root node $t$ and a given feature vector $\phi$, the corresponding formula $\varphi_t$ is recursively defined as follows:

$$\varphi_t = \bot \vee \bigvee_{t_{\text{child}} \in t} \Bigg( (\phi^j < L_{t_{\text{child}}}) \vee (\phi^j > R_{t_{\text{child}}})$$
$$\vee \left( (\phi^j \geq L_{t_{\text{child}}}) \wedge (\phi^j \leq R_{t_{\text{child}}}) \wedge \varphi_{t_{\text{child}}} \right) \Bigg). \quad (2)$$

In this context, $j$ denotes the split dimension corresponding to node $t$. Note that $\varphi_t$ evaluates to $\bot$ if $t$ is a leaf node. When $\varphi$ evaluates to $\top$, it signifies that the input feature vector corresponds to a state-action tuple that violates the constraints. For the simple navigation examples the following formula can be extracted from the expert tree presented in Figure 2:

$$\varphi = (\phi^0 < 0.1) \vee (\phi^0 > 0.9)$$
$$\vee \left( (\phi^0 > 0.1) \wedge (\phi^0 < 0.7) \wedge (\phi^1 < 0.1) \right)$$
$$\vee \left( (\phi^0 > 0.1) \wedge (\phi^0 < 0.7) \wedge (\phi^1 > 0.3) \right)$$
$$\vee \left( (\phi^0 > 0.7) \wedge (\phi^0 < 0.9) \wedge (\phi^1 < 0.1) \right)$$
$$\vee \left( (\phi^0 > 0.7) \wedge (\phi^0 < 0.9) \wedge (\phi^1 > 0.9) \right).$$

Additionally, for each dimension $j$, two more rules are incorporated to invalidate feature representations where $\phi^j$ is lower than the observed minimum in the expert trajectories: $\min \left( \phi^j \; : \; \forall \phi \in \mathcal{D} \right)$ or higher than the observed maximum: $\max \left( \phi^j \; : \; \forall \phi \in \mathcal{D} \right)$.

## Constraint-Based Cost Function and Optimization

The extracted formula $\varphi$ gives rise to a cost function that can be utilized within a CRL framework. In cases where a constraint is violated, the agent incurs a cost of 1; conversely, if all constraints are adhered to, the cost remains at 0. To tackle this constrained problem, we employ the Lagrangian method in conjunction with Proximal Policy Optimization (PPO) (Schulman et al. 2017). This constrained problem can then be addressed as an unconstrained max-min optimization problem, and we employ the implementation provided by the Omnisafe framework (Ji et al. 2023). It's noteworthy that the PPO-Lagrangian approach, while conservative, has been shown to yield comparable or superior results to other methods such as Constrained Policy Optimization (CPO) (Achiam et al. 2017), Projected Constrained Policy Optimization (PCPO) (Yang et al. 2020), and First-Order Constrained Policy Optimization with Penalty (FOCOPS) (Zhang, Vuong, and Ross 2020) on various safety gym benchmark environments, as discussed by Ji et al. (2023) and Ray, Achiam, and Amodei (2019).

## Refining Constraint Definitions

It is important to mention that the learned tree could serve directly as a cost function. However, if our intention is to prune the set of constraints to improve interpretability, the DNF description of the constraints becomes more practical. We interpret each conjunction in $\varphi$ as an individual rule or constraint. During the training of the CRL agent, we maintain a record of the number of times each conjunction is evaluated and how many times it evaluates to true (i.e. the corresponding constraint is violated). Conjunctions that exhibit a violation-evaluation ratio below a predefined threshold are pruned from the constraint definition. The rationale behind this approach lies in the fact that during training, the learning agent strives to maximize the discounted sum of rewards. When rules are violated during the execution of highly rewarding behavior, it suggests that these rules likely correspond to genuine constraints since the expert agent actively avoided these states. Conversely, when rules are never violated, they are less likely to be true constraints, as they pertain to regions in the feature space that are scarcely visited by the learning agent, indicating low reward potential.

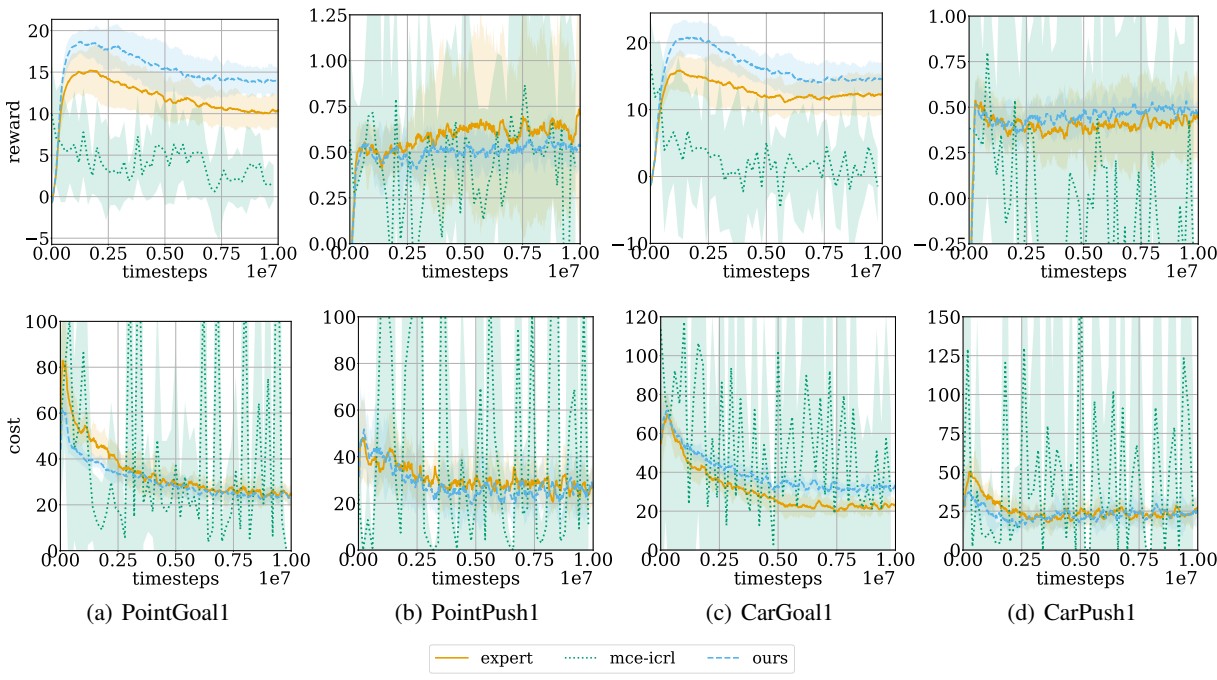

Figure 3: Reward (top) and ground truth cost (bottom) during training of agents in the synthetic benchmark environments.

When evaluating a feature representation, the conjunctions are assessed in ascending order of complexity. As a result, conjunctions with fewer literals are given preference over more intricate rules. For the simple navigation example, the pruned constraints are defined as follows:

$$\varphi = (\phi^0 > 0.1) \wedge (\phi^0 < 0.7) \wedge (\phi^1 > 0.3).$$

It is worth highlighting that if we examine the trajectories taken by an initial learning agent, as depicted by the dashed lines in Figure 1, we will observe a notably high violation-evaluation ratio for this rule. On the contrary the rules which are pruned describe areas within the feature space where the potential rewards are limited, and consequently, these regions are seldom explored by the learning agent.

## Results

We evaluate our method on a set of synthetic safe RL benchmark domains proposed by Ray, Achiam, and Amodei (2019) and a realistic highway environment (Krajewski et al. 2018). All results are averaged over 10 trials with random seeds. In the appendix, we provide examples of the learned formulas.

### Synthetic Environments

We assess two categories of agents: The *Point* agent, a basic robot confined to a 2D plane, equipped with two actuators for rotation and forward/backward movement, and the *Car* agent, a somewhat more complex robot with two independently driven parallel wheels and a free-rolling rear wheel. In the case of the *Car* robot, both steering and forward/backward movement necessitate coordination between the two

drives. Our evaluation encompasses two distinct tasks: *Goal* and *Push*. In the *Goal* task, the agent's aim is to reach a specified goal location while circumventing hazards. The *Push* task involves the agent pushing a box to the designated goal position while avoiding hazards. The feature space for the various agents is characterized by the agent's acceleration and velocity along both the x and y axes, in addition to the data from 16 lidar sensors: $a_x, a_y, v_x, v_y, d_{l0:15}$. The lidar sensor readings indicate the distance in each of the 16 directions around the agent to a hazard. The lidar values range from 0 to 1, with a reading of 1 signifying that the agent is in a hazardous area.

Figure 3 illustrates the acquired rewards and costs during the training of different agents. These agents include one provided with ground truth constraints and trained using PPO-Lagrangian (i.e. expert), another trained with expert demonstrations using Maximum Causal Entropy Inverse Constrained Reinforcement Learning (MCE-ICRL) (Baert et al. 2023) as a state-of-the-art inverse constrained reinforcement learning method, and an agent trained with expert demonstrations using our proposed method (i.e. ours). We leverage the expert agent's policy, trained using the ground truth constraints, to sample trajectories, which serve as input expert trajectories for both MCE-ICRL and our method. For both the *Goal* and *Push* tasks our method approximates the reward and cost achieved by the expert. The *Push* task presents a significantly greater challenge compared to the *Goal* task, as is reflected by the lower rewards obtained. Even when provided with ground truth constraints, current CRL methods struggle to learn a satisfactory policy. Consequently, the expert trajectories will be sub-optimal, in-

| source/target | PointGoal1 | PointPush1 | CarGoal1 | CarPush1 |
|---|---|---|---|---|
| PointGoal1 | $13.9 \pm 1.7$ | $0.6 \pm 0.3$ | $4.9 \pm 2.1$ | $0.7 \pm 0.3$ |
| PointPush1 | $0.5 \pm 0.2$ | $0.6 \pm 0.1$ | $0.1 \pm 0.2$ | $0.4 \pm 0.1$ |
| CarGoal1 | $12.2 \pm 3.6$ | $0.9 \pm 0.3$ | $15.1 \pm 3.1$ | $0.7 \pm 0.2$ |
| CarPush1 | $0.1 \pm 0.1$ | $0.5 \pm 0.1$ | $0.3 \pm 0.3$ | $0.5 \pm 0.1$ |

Table 1: Transferring constraints between agents and tasks: reward

| source/target | PointGoal1 | PointPush1 | CarGoal1 | CarPush1 |
|---|---|---|---|---|
| PointGoal1 | $27.9 \pm 5.0$ | $54.5 \pm 16.7$ | $35.1 \pm 11.6$ | $47.4 \pm 9.3$ |
| PointPush1 | $4.5 \pm 1.7$ | $26.2 \pm 7.3$ | $6.5 \pm 3.9$ | $18.7 \pm 8.2$ |
| CarGoal1 | $41.0 \pm 8.0$ | $50.3 \pm 12.8$ | $33.7 \pm 5.3$ | $48.7 \pm 9.9$ |
| CarPush1 | $13.9 \pm 4.4$ | $25.7 \pm 8.3$ | $14.4 \pm 3.5$ | $24.6 \pm 10.6$ |

Table 2: Transferring constraints between agents and tasks: ground truth cost

evitably impacting the performance of our method, as these trajectories serve as its fundamental input. In essence, the efficacy of our method is upper bounded by the quality of the expert demonstrations. Furthermore, the same oracle CRL method is part of our proposed approach, which raises concerns that its failure in cases where ground truth constraints are available may also extend to situations where it operates with learned constraints. It is notable that MCE-ICRL fails to recover a satisfactory policy in our experiments. This can be attributed to the complexity of the environments which significantly surpasses those utilized in the original paper (Baert et al. 2023).

One key advantage of defining or learning constraints is that they often are the same for various agents and tasks. This is beneficial because we can learn the constraints once and use them in multiple settings. To this end, we assess to what extent the constraints learned from demonstrations of an agent of type A performing task X can be transferred to agents of type B performing task Y. In table 1 and 2, we present the reward and cost, respectively, acquired by agents after training. The rows indicate the environment in which the rules are learned (source domain) and the columns correspond with the environment in which the agent is trained (target domain). The main diagonal contains the results when the source and target domain are the same. The best results for a given target domain are achieved when the source and target domains are the same. Notably, the most favorable outcomes are observed when transferring constraints between different agents performing the same task (e.g. *CarGoal* to *Point-Goal*). However, when constraints are transferred between distinct tasks for the same agent, there is a noticeable decline in performance (e.g. *PointPush* to *PointGoal*). Analyzing the learned formula from expert trajectories performing the *Push* task, we find that more restrictive constraints are learned compared to when expert trajectories optimize the *Goal* task. This discrepancy is reflected in the results, where employing constraints learned in the *Goal* domain within the *Push* domain leads to a higher frequency of constraint violations, suggesting that the constraints are excessively lenient. Conversely, employing constraints acquired in the *Push* domain within the *Goal* domain results in lower costs but also reduced rewards, indicating that the constraints are overly

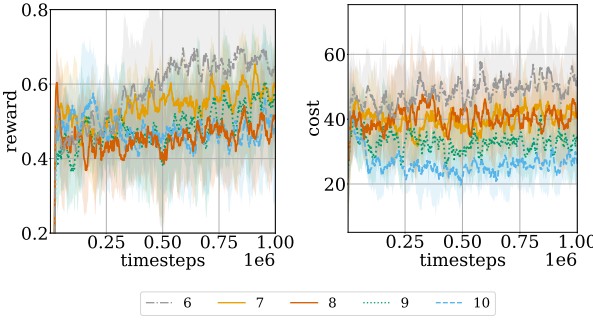

Figure 4: Reward (left) and cost (right) during training in the *CarPush* environment for constraints extracted from trees with various depths.

restrictive.

For each environment, we determine the ideal tree depth for achieving optimal results. It's important to note that as the tree depth increases, the learned formula becomes more restrictive. A more restrictive formula used as a cost function tends to produce trajectories that closely resemble expert trajectories, often leading to lower overall costs. However, it's essential to consider that in such scenarios, there is a risk of overfitting the learned tree to the training data. This can result in a situation where the model does not allow for regions within the feature space which are not observed in the expert trajectories but not necessarily invalid. This could limit the agent's exploration and generalization capabilities and consequently may lead to lower rewards. This reasoning is exemplified in Figure 4 where we depict the acquired reward and cost by agents trained on cost functions originating from trees with various depths.

## Realistic Highway Environment

Traffic is a real-world environment where agent behavior is heavily governed by both explicit and implicit rules. We extract expert demonstrations from the highD dataset (Krajewski et al. 2018), a comprehensive repository of annotated vehicle trajectories recorded on German high-

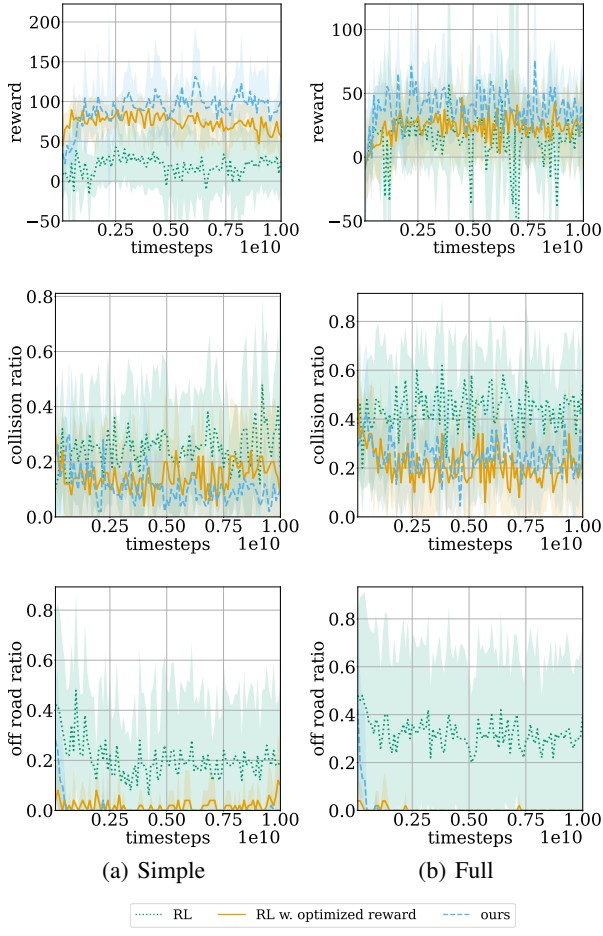

(a) Simple          (b) Full

······ RL     —— RL w. optimized reward     --- ours

Figure 5: Reward (top), collision rate (middle) and off road ratio (bottom) during training in a realistic highway environment.

ways. To facilitate the training of Reinforcement Learning (RL) agents within this environment, we transform this dataset into multiple scenarios adhering to the Common-Road framework (Althoff, Koschi, and Manzinger 2017). Subsequently, we employ the CommonRoad-RL framework for agent training (Wang, Krasowski, and Althoff 2021). All the scenarios derived from the highD dataset encompass either two or three lanes in both directions. The feature space is characterized by the velocity along the x- and y-axes, the relative position of agents leading and following the ego-agent in the left, same and right lane, and the proximity to the left and right road edges $\{v_x, v_y, p_{\text{rel}_0}, p_{\text{rel}_1}, p_{\text{rel}_2}, p_{\text{rel}_3}, p_{\text{rel}_4}, p_{\text{rel}_5}, d_{\text{left}}, d_{\text{right}}\}$. The output of the distance sensor, which gauges the distance to surrounding vehicles, spans a range from 0 to 500. Our experiments entail two distinct settings: one where an agent is trained on a restricted set of 52 scenarios from the highD dataset (i.e. highD simple) and another where an agent is trained on all 3000 scenarios (i.e. highD full). Due to the absence of ground truth constraints in this environment, it is not feasible to report the frequency of constraint violations. Instead, we present two key performance metrics: the off-road ratio (indicating the percentage of trajectories in which the agent deviates from the road) and the collision ratio (reflecting the percentage of trajectories in which the agent collides with another vehicle). For the purpose of comparison, our approach is evaluated against two other agent types: a standard RL agent without any provided constraints (i.e. RL) and an RL agent equipped with an engineered reward function, as proposed by Wang, Krasowski, and Althoff (2021), which issues negative rewards when the agent deviates off-road or collides with another vehicle (i.e. RL w. optimized reward). The results displayed in Figure 5 clearly demonstrate that our method excels in preventing agents from driving off-road and reducing the incidence of collisions with other vehicles. These outcomes strongly suggest that our rule-learned agents are not only safer but also achieve higher rewards compared to both the conventional RL agents and those equipped with the hand-engineered optimized reward function.

## Related Work

The constraint learning problem is by definition ill-posed as many set of constraints can describe the same set of expert trajectories. In order to prevent the acquisition of overly cautious constraints, many approaches rely on the principle of maximum entropy to derive the most concise set of constraints that align with expert demonstrations (Ziebart et al. 2008). Scobee and Sastry (2020) introduced an approach focused on learning a set of constraints that maximize the likelihood of expert demonstrations. This method, when combined with inductive logic programming, allows for the acquisition of a logical formula representing the learned constraints (Baert, Leroux, and Simoens 2023). By approximating the maximum likelihood objective, this approach can be effectively applied to environments with continuous state-action spaces (Malik et al. 2021; Gaurav et al. 2023; Liu et al. 2023). However, it's important to emphasize that these methods are primarily well-suited for environments with deterministic dynamics. For environments with stochastic dynamics, constraint learning is made possible through methods based on the principle of maximum causal entropy (Ziebart, Bagnell, and Dey 2010), as demonstrated by McPherson, Stocking, and Sastry (2021) for discrete environments and by Baert et al. (2023) for continuous environments. Unlike our methodology, the aforementioned methods all necessitate knowledge of the agent's goal and the availability of a simulator of the environment for learning the constraints. Of particular relevance to our research is the work of Lindner et al. (2023), who also define constraints as a convex set in the environment's feature vector space. Nevertheless, their definition of the safe set involves the convex hull of the feature expectations derived from the demonstrations. In contrast, our approach models the safe set using a decision tree, offering the distinct advantage of interpretability for both the safe set and the constraints extracted. Additionally, we hypothesize that our method is less prone to overfitting, as it employs only the most informative feature dimensions for defining the safe set. Furthermore,

it is important to note that they learn a linear cost function in feature space. To allow their method to work in environments with non-linear constraints, they adopt a one-hot encoding of the state-action space as features. The latter is only possible when the state-action space is relatively small and finite. Because of this the method proposed by Lindner et al. (2023) is unsuitable for learning constraints in the environments we evaluated. Another benefit of our method is that the OC-tree naturally provides robustness when dealing with a limited number of negative examples in the training data, a common scenario when learning from human demonstrations.

## Conclusion

We have introduced a novel approach to acquire constraints modeled by a logic formula in disjunctive normal form from expert demonstrations. These acquired rules can be employed as an easily interpretable cost function in the context of constrained reinforcement learning. We evaluated our method on multiple synthetic environments and realistic autonomous driving scenarios. In both, our method improves safety and performance of the learning agents. In addition, we provided a post-training pruning mechanism to simplify to learned formula. Nevertheless, there exists substantial potential for enhancing the transferability of constraints across various domains. This improvement will alleviate the necessity for expert trajectories to be available in every target domain. In future work, we plan to investigate the relation between the tree depth and the transferability of the learned constraints. We would also like to investigate whether more effective constraints can be learned when a simulator of the environment is provided. At last, we want to extend the proposed method for learning constraints expressed in temporal logic. This extension brings the significant benefit of enhanced expressiveness, enabling us to effectively describe more intricate and complex constraints.

## Examples of Learned Rules

We present some learned formulas describing the constraints in an environment. The presented formulas are already pruned using a violation-evaluation threshold of $0.001$.
**PointGoal**: The first rules define a maximum acceleration and velocity. The following rules define a safe boundary between the agent and hazards in different directions around the agent.
**highD**: The first rules define a minimum and maximum speed limit along the x- and y-axis and a limit on the distance between the car in the same lane in front of the agent ($p_{\mathrm{rel}_4}$) and behind the agent ($p_{\mathrm{rel}_1}$). The next rules define a minimum distance to the road edges. The following rules distinguish the agent being on the left or right lane based on the distance to the right road edge. The last rule states the distance to the vehicle on the left lane in front of the agent should be smaller than 344.91 if the distance to the vehicle on the left lane behind the agent is bigger than 325.83. This will force the agent to move one lane to the left. It is debatable whether this last rule corresponds to desired behavior as the agent will have a preference to drive on the left or middle lane.

## Acknowledgement

This research was partially funded by the Flemish Government (Flanders AI Research Program).

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

Rules extracted from the expert trajectories in the *PointGoal* environment after pruning.

$$\varphi = a_x < -5.42 \vee v_y < -1.05 \vee v_x > 1.46 \vee d_{l0} > 0.88 \vee d_{l3} > 1.0 \vee d_{l5} > 0.99$$
$$\vee\, d_{l0} >= 0.0 \wedge d_{l0} <= 0.22 \wedge d_{l1} > 0.81$$
$$\vee\, d_{l0} >= 0.22 \wedge d_{l0} <= 0.88 \wedge d_{l7} > 0.87$$
$$\vee\, d_{l0} >= 0.0 \wedge d_{l0} <= 0.22 \wedge d_{l1} >= 0.0 \wedge d_{l1} <= 0.27 \wedge d_{l15} > 0.66$$
$$\vee\, d_{l0} >= 0.0 \wedge d_{l0} <= 0.22 \wedge d_{l1} >= 0.27 \wedge d_{l1} <= 0.81 \wedge d_{l15} > 0.81$$
$$\vee\, d_{l0} >= 0.22 \wedge d_{l0} <= 0.88 \wedge d_{l7} >= 0.0 \wedge d_{l7} <= 0.27 \wedge d_{l6} > 0.85$$
$$\vee\, d_{l0} >= 0.22 \wedge d_{l0} <= 0.88 \wedge d_{l7} >= 0.27 \wedge d_{l7} <= 0.87 \wedge d_{l14} > 0.94$$
$$\vee\, d_{l0} >= 0.0 \wedge d_{l0} <= 0.22 \wedge d_{l1} >= 0.0 \wedge d_{l1} <= 0.27 \wedge d_{l15} >= 0.0 \wedge d_{l15} <= 0.66 \wedge d_{l2} > 0.83$$
$$\vee\, d_{l0} >= 0.0 \wedge d_{l0} <= 0.22 \wedge d_{l1} >= 0.27 \wedge d_{l1} <= 0.81 \wedge d_{l15} >= 0.0 \wedge d_{l15} <= 0.81 \wedge d_{l14} > 0.94$$
$$\vee\, d_{l0} >= 0.22 \wedge d_{l0} <= 0.88 \wedge d_{l7} >= 0.0 \wedge d_{l7} <= 0.27 \wedge d_{l6} >= 0.0 \wedge d_{l6} <= 0.31 \wedge d_{l8} > 0.78$$
$$\vee\, d_{l0} >= 0.22 \wedge d_{l0} <= 0.88 \wedge d_{l7} >= 0.0 \wedge d_{l7} <= 0.27 \wedge d_{l6} >= 0.31 \wedge d_{l6} <= 0.85 \wedge d_{l8} > 0.89$$
$$\vee\, d_{l0} >= 0.22 \wedge d_{l0} <= 0.88 \wedge d_{l7} >= 0.27 \wedge d_{l7} <= 0.87 \wedge d_{l14} >= 0.0 \wedge d_{l14} <= 0.28 \wedge d_{l13} > 0.94$$
$$\vee\, d_{l0} >= 0.22 \wedge d_{l0} <= 0.88 \wedge d_{l7} >= 0.27 \wedge d_{l7} <= 0.87 \wedge d_{l14} >= 0.28 \wedge d_{l14} <= 0.94 \wedge d_{l4} > 0.94$$

Rules extracted from the human trajectories in the highD dataset after pruning.

$$\varphi = v_y < -0.95 \vee v_y > 0.96 \vee v_x < 20.67 \vee v_x > 47.47 \vee p_{\mathrm{rel}_1} < 18.83 \vee p_{\mathrm{rel}_4} < 18.83 \vee d_{\text{left edge}} < 4.42 \vee d_{\text{right edge}} < 2.02$$
$$\vee\, v_y >= -0.95 \wedge v_y <= 0.96 \wedge d_{\text{right edge}} >= 4.65 \wedge d_{\text{right edge}} <= 7.38 \wedge p_{\mathrm{rel}_1} >= 18.83 \wedge p_{\mathrm{rel}_1} <= 328.76 \wedge p_{\mathrm{rel}_0} < 236.11$$
$$\vee\, v_y >= -0.95 \wedge v_y <= 0.96 \wedge d_{\text{right edge}} >= 4.65 \wedge d_{\text{right edge}} <= 7.38 \wedge p_{\mathrm{rel}_1} >= 328.76 \wedge p_{\mathrm{rel}_1} <= 500.0 \wedge p_{\mathrm{rel}_0} < 155.9$$
$$\vee\, v_y >= -0.95 \wedge v_y <= 0.96 \wedge d_{\text{right edge}} >= 4.65 \wedge d_{\text{right edge}} <= 7.38 \wedge p_{\mathrm{rel}_1} >= 18.83 \wedge p_{\mathrm{rel}_1} <= 328.76 \wedge p_{\mathrm{rel}_0} >= 236.11$$
$$\wedge\, p_{\mathrm{rel}_0} <= 500.0 \wedge p_{\mathrm{rel}_3} < 235.66$$
$$\vee\, v_y >= -0.95 \wedge v_y <= 0.96 \wedge d_{\text{right edge}} >= 4.65 \wedge d_{\text{right edge}} <= 7.38 \wedge p_{\mathrm{rel}_1} >= 328.76 \wedge p_{\mathrm{rel}_1} <= 500.0 \wedge p_{\mathrm{rel}_0} >= 155.9$$
$$\wedge\, p_{\mathrm{rel}_0} <= 500.0 \wedge p_{\mathrm{rel}_3} < 136.43$$
$$\vee\, v_y >= -0.95 \wedge v_y <= 0.96 \wedge d_{\text{right edge}} >= 2.02 \wedge d_{\text{right edge}} <= 4.65 \wedge p_{\mathrm{rel}_0} >= 0.04 \wedge p_{\mathrm{rel}_0} <= 325.83$$
$$\wedge\, p_{\mathrm{rel}_3} >= 0.03 \wedge p_{\mathrm{rel}_3} <= 332.09 \wedge p_{\mathrm{rel}_4} < 26.81$$
$$\vee\, v_y >= -0.95 \wedge v_y <= 0.96 \wedge d_{\text{right edge}} >= 2.02 \wedge d_{\text{right edge}} <= 4.65 \wedge p_{\mathrm{rel}_0} >= 325.83 \wedge p_{\mathrm{rel}_0} <= 500.0 \wedge p_{\mathrm{rel}_1} >= 33.84$$
$$\wedge\, p_{\mathrm{rel}_1} <= 219.6 \wedge p_{\mathrm{rel}_4} < 29.48$$
$$\vee\, v_y >= -0.95 \wedge v_y <= 0.96 \wedge d_{\text{right edge}} >= 2.02 \wedge d_{\text{right edge}} <= 4.65 \wedge p_{\mathrm{rel}_0} >= 325.83 \wedge p_{\mathrm{rel}_0} <= 500.0 \wedge p_{\mathrm{rel}_1} >= 219.6$$
$$\wedge\, p_{\mathrm{rel}_1} <= 500.0 \wedge p_{\mathrm{rel}_3} > 344.91$$

McPherson, D. L.; Stocking, K. C.; and Sastry, S. S. 2021. Maximum likelihood constraint inference from stochastic demonstrations. In *2021 IEEE Conference on Control Technology and Applications (CCTA)*, 1208–1213. IEEE.

Ray, A.; Achiam, J.; and Amodei, D. 2019. Benchmarking safe exploration in deep reinforcement learning. *arXiv preprint arXiv:1910.01708*, 7(1): 2.

Russell, S. 2019. *Human compatible: Artificial intelligence and the problem of control*. Penguin.

Schulman, J.; Wolski, F.; Dhariwal, P.; Radford, A.; and Klimov, O. 2017. Proximal Policy Optimization Algorithms. *CoRR*, abs/1707.06347.

Scobee, D. R. R.; and Sastry, S. S. 2020. Maximum Likelihood Constraint Inference for Inverse Reinforcement Learning. In *8th International Conference on Learning Representations, ICLR 2020, Addis Ababa, Ethiopia, April 26-30, 2020*. OpenReview.net.

Wang, X.; Krasowski, H.; and Althoff, M. 2021. CommonRoad-RL: A Configurable Reinforcement Learning Environment for Motion Planning of Autonomous Vehicles. In *IEEE International Conference on Intelligent Transportation Systems (ITSC)*.

Yang, T.; Rosca, J.; Narasimhan, K.; and Ramadge, P. J. 2020. Projection-Based Constrained Policy Optimization. In *8th International Conference on Learning Representations, ICLR 2020, Addis Ababa, Ethiopia, April 26-30, 2020*. OpenReview.net.

Zhang, Y.; Vuong, Q.; and Ross, K. 2020. First order constrained optimization in policy space. *Advances in Neural Information Processing Systems*, 33: 15338–15349.

Ziebart, B. D.; Bagnell, J. A.; and Dey, A. K. 2010. Modeling Interaction via the Principle of Maximum Causal Entropy. In Fürnkranz, J.; and Joachims, T., eds., *Proceedings of the 27th International Conference on Machine Learning (ICML-10), June 21-24, 2010, Haifa, Israel*, 1255–1262. Omnipress.

Ziebart, B. D.; Maas, A. L.; Bagnell, J. A.; Dey, A. K.; et al. 2008. Maximum entropy inverse reinforcement learning. In *Aaai*, volume 8, 1433–1438. Chicago, IL, USA.