# OpenReview forum: "Learning Safety Constraints From Demonstration Using One-Class Decision Trees"
_AAAI.org/2024/Workshop/NuCLeaR — NuCLeaR 2024_

### Official Review · Reviewer_5iDF · 2023-12-06
**Good paper with a new one-class DT to learn constraints**

**Rating:** 7
**Confidence:** 5

**Review:**

This is a well written paper with clear explanations on the problem context, related work and background. Learning interpretable constraints from demonstrations to improve safety of RL agents is valuable but under-explored. This paper proposes a one-class decision trees to learn safe convex set in feature space. The implementation steps are clearly presented. The test case in autonomous driving simulations demonstrate applicability to complex, safety-critical real world tasks. In additional, the transfer learning experiments provide some evidence on the generalizability of learned constraints beyond isolated tasks.

There are a few limitations in the paper:
1) Constraint learning quality heavily depends on expert demonstration quality and size of the data. This could result in overfitting and bias in the learned constraints for the current few thousand examples used.
2) More analysis is needed for generalization across tasks in terms of transferability of learned constraints, as even slightly different tasks can degrade performance, implying there might be exploiting spurious patterns.
3) there is no human evaluation or visual assessment on whether learned constraints actually capture meaningful safe behavior.
4) only simple simulation environments and one driving dataset are used for evaluation.
5) No comparison of constraint quality with other constraint /rule inference methods limits the implied benefits of proposed one-class decision tree approach

---

### Official Review · Reviewer_BqFo · 2023-12-07
**The paper explores the use of one class decision trees to acquire interpretable safety constraints for CRL applications. while promising further analysis, benchmarking, and demonstration quality improvements could better address limitations and facilitate real-world use.**

**Rating:** 7
**Confidence:** 3

**Review:**

The paper presents a novel method for learning interpretable safety constraints from expert demonstrations using one-class decision trees. Pros of the approach include
- Interpretability of the learned logical constraints
- The ability to reduce manual specification of constraints
- Improved safety and performance over baselines in experiments

However, some limitations exist as well:
- The approach relies heavily on demonstration quality
- The tree depth likely requires careful tuning for each environment to prevent overfitting while ensuring reasonable performance
- Needs more analysis on transferability of learned constraints, which the authors have mentioned as a future research direction

---

### Decision · Program_Chairs · 2023-12-11

Accept